# The Effect of RBP4 on microRNA Expression Profiles in Porcine Granulosa Cells

**DOI:** 10.3390/ani11051391

**Published:** 2021-05-13

**Authors:** Yun Zhao, Jiahui Rao, Tong Qiu, Chunjin Li, Xu Zhou

**Affiliations:** 1College of Animal Sciences, Jilin University, Changchun 130062, China; zhao_yun@jlu.edu.cn; 2College of Veterinary Medicine, Jilin University, Changchun 130062, China; raojh@jlu.edu.cn (J.R.); qt0821@foxmail.com (T.Q.)

**Keywords:** granulosa cells (GCs), microRNAs, retinol binding protein 4 (*RBP4*)

## Abstract

**Simple Summary:**

Retinol binding protein 4 (RBP4), mainly secreted by the liver and adipocytes, is a transporter of vitamin A. RBP4 has been shown to be involved in several pathophysiological processes, such as polycystic ovary syndrome (PCOS), obesity, insulin resistance, and cardiovascular risk. However, the role of RBP4 in mammalian follicular granulosa cells (GCs) remains largely unknown. To characterize the molecular pathways associated with the effects of RBP4 on GCs, we used sRNA deep sequencing to detect differential microRNA (miRNA) expression in GCs overexpressing RBP4. A total of 17 miRNAs were significantly different between the experimental and control groups. Our results support the notion that several miRNAs are involved in important biological processes associated with folliculogenesis and pathogenesis. These results will be useful for further studies investigating the role of RBP4 in porcine GCs.

**Abstract:**

Retinol binding protein 4 (RBP4) is a transporter of vitamin A that is secreted mainly by hepatocytes and adipocytes. It affects diverse pathophysiological processes, such as obesity, insulin resistance, and cardiovascular diseases. MicroRNAs (miRNAs) have been reported to play indispensable roles in regulating various developmental processes via the post-transcriptional repression of target genes in mammals. However, the functional link between RBP4 and changes in miRNA expression in porcine granulosa cells (GCs) remains to be investigated. To examine how increased expression of *RBP4* affects miRNA expression, porcine GCs were infected with *RBP4*-targeted lentivirus for 72 h, and whole-genome miRNA profiling (miRNA sequencing) was performed. The sequencing data were validated using real-time quantitative polymerase chain reaction (RT-qPCR) analysis. As a result, we obtained 2783 known and 776 novel miRNAs. In the experimental group, 10 and seven miRNAs were significantly downregulated and upregulated, respectively, compared with the control group. Ontology analysis of the biological processes of these miRNAs indicated their involvement in a variety of biological functions. Gene Ontology and Kyoto Encyclopedia of Genes and Genomes enrichment analyses indicated that these miRNAs were involved mainly in the chemokine signaling pathway, peroxisome proliferators-activated receptors (PPAR) signaling pathway, insulin resistance pathway, nuclear factor-kappa B(NF-kappa B) signaling pathway, and steroid hormone biosynthesis. Our results indicate that RBP4 can regulate the expression of miRNAs in porcine GCs, with consequent physiological effects. In summary, this study profiling miRNA expression in *RBP4*-overexpressing porcine GCs provides an important reference point for future studies on the regulatory roles of miRNAs in the porcine reproductive system.

## 1. Introduction

Despite the hundreds of porcine follicles whose growth begins at estrus, only 10–20 follicles are selected to release a mature oocyte [1]. Folliculogenesis involves many events at the cellular and molecular levels that occur in a highly orchestrated manner, culminating in ovulation [2]. During this process, bidirectional communication between oocytes and GCs is essential for follicular development and oocyte growth. Numerous studies have elucidated the varied functions of GCs in the formation of ovarian follicles [3,4]. This evidence illustrates that endocrine/paracrine/autocrine factors such as gonadotropins, growth factors, and the ovarian microenvironment and several signaling pathways have effects on the differentiation of GCs [5,6,7,8]. However, an understanding of alterations in gene expression during GC differentiation and the function of factors that are specifically expressed in GCs is still lacking.

RBP4, known as a retinol transporter, a member of the lipocalin family [9], is synthesized and secreted mainly in the liver, although this protein is expressed in other tissues (including adipose tissue, kidney, skeletal muscle, ovary, and testis [10,11,12,13]) at low levels. In humans, RBP4 levels in serum have been found to be related to many metabolic diseases, such as insulin resistance, type 2 diabetes mellitus, and obesity [14,15]. Recently, many studies have demonstrated the involvement of RBP4 in female reproduction. In humans, serum RBP4 levels decrease with diminishing ovarian reserve and correlate positively with chronic inflammation-related components, including increased serum high sensitivity C-reactive protein and anti-Mullerian hormone levels [16]. Additionally, RBP4, identified as a serum marker for ovarian cancer, directly induces cancer progression of ovarian cancer cells by increasing the expression of the cancer metastasis factors *MM2* and *MM9* through the RhoA/Rock1 pathway [17,18]. Serum concentrations of RBP4 are increased in women with polycystic ovary syndrome (PCOS) and inadequately high concentrations of androgen and insulin resistance are characteristic of PCOS [19,20,21]. Additionally, *RBP4* expression in adipose tissues is regulated by 17-estradiol, and the levels of RBP4 in serum are elevated in women with PCOS and obesity [19]. These data suggest that RBP4 might be involved in normal ovarian function.

In recent years, a class of non-coding, single-stranded small RNAs (microRNAs (miRNAs)) have been suggested to have regulatory functions during post-transcriptional stages in many physiological processes [22]. miRNAs play an important role in the active regulation of cellular and biological processes, such as cell migration [23,24], apoptosis [25], proliferation [26,27], and metabolism [28]. Multiple studies have demonstrated that aberrant miRNA expression in female reproductive tissues/cells and embryos is related to disordered folliculogenesis and GC proliferation [29,30]. miRNAs regulate the function of GCs by altering the expression levels of target genes. For example, in the granulosa-like tumor cell line, miR-1388 promotes cell cycle progression by increasing *cyclin D1* and decreasing *p21* levels by targeting potassium voltage-gated channel subfamily A member 5 (*KCNA5*) [31]. Another study showed that miR-29c participates in the regulation of follicular atresia by inhibiting GC apoptosis by targeting *SMARCA2* via the NORFA-SMAD4 axis [32].

In previous studies in our lab, Sun et al. [33] detected a high level of RBP4 in the follicular fluids of cystic follicles. We observed that RBP4 could stimulate the expression of follicle-stimulating hormone receptor (*FSHR*) and luteinizing hormone receptor (*LHR*) in GCs cultured in vitro [34] and identified the effects of RBP4 on GCs by transcriptomic analysis following *RBP4* overexpression [35]. Taken together, these results show that there is a high level of RBP4 expression in follicular fluids in cystic follicles, which led us to assume the possible association of RBP4 with PCOS and cystic follicles. To date, the specific effects of different miRNAs and the mechanisms, particularly the target genes and signaling pathways, by which they regulate GCs treated with RBP4 are poorly defined.

## 2. Materials and Methods

### 2.1. Obligatory Ethical Approval

All experiments in the present study were performed according to the instructions of the Animal Care and Use Committee of Jilin University Changchun, China (approval number: SY201912003).

### 2.2. Culture and Treatment of GCs

Landrace porcine ovaries obtained from sows aged 165–180 days from a local slaughter abattoir (Changchun Huazheng Production Cooperation, Changchun, Jilin, China) were transported to the laboratory within 20 min and maintained at 37 °C for isolation of GCs. The follicular fluid was aspirated aseptically from 3–6 mm antral follicles using a 10 mL syringe, centrifuged at 500× *g* for 5 min, and rinsed twice with preheated phosphate-buffered saline (PBS) (Hyclone, Logan, UT, USA) according to previously described methods [36]. GCs were seeded at an initial density of 1 × 10^6^ cells/mL in six-well plates (Corning Inc., Shanghai, China) and cultured in Dulbecco’s modified Eagle’s medium/Ham’s F-12 (DMEM/F12) (Hyclone, Logan, UT, USA) supplemented with 10% fetal bovine serum (FBS) (Invitrogen, Carlsbad, CA, USA) and 1% penicillin–streptomycin (Hyclone, Logan, UT, USA) in a humidified incubator with 5% CO_2_ at 37 °C.

A pLVX-IRES-ZsGreen-*RBP4* plasmid was constructed by inserting the *RBP4* gene of 606 bp (GenBank Accession No. NM_214057.1) into the pLVX-ZsGreen vector (Invitrogen, Carlsbad, CA, USA) at the XhoI and NotI sites. To produce lentivirus, FuGENE HD reagent (Roche, Mannheim, Germany) was used to transfect 293T cells (ATCC, CRL-11268; Manassas, VA, USA) with the combined packaging system (Invitrogen, Carlsbad, CA, USA), including a vector plasmid, the structure plasmids PLP1 and PLP2, and the envelope plasmid VSVG, following the manufacturer’s instructions. Lentiviral particles were harvested from the supernatant 48 and 72 h after transfection and purified by ultracentrifugation. 293T cells transfected with the pLVX-IRES-ZsGreen blank vector were used as negative controls. The lentiviral particles were added to the GCs at a multiplicity of infection of 50 for 72 h. GCs were divided into the following two experimental groups: control group (pLVX-IRES-ZsGreen-infected cells) and RBP4 group (pLVX -IRES-ZsGreen-*RBP4*-infected cells). RT-qPCR and Western blotting were used to confirm the success of RBP4 expression modification in GCs (the detailed infected porcine GC data are shown in another published paper [35]). Each experiment was performed in biological triplicate.

### 2.3. Library Construction, miRNA Sequencing, and Data Analysis

Total RNA was isolated from GCs using a protocol based on TRIzol reagent (Takara, Dalian, China). RNA concentration and purity were assessed using a Nanodrop 2000 spectrophotometer (Thermo Fisher, Waltham, MA, USA); the OD260/OD280 value needed to be higher than 1.8. RNA integrity was assessed by agarose gel electrophoresis. Six small RNA-sequencing (sRNA-seq) libraries (three sRNA-seq libraries for the RBP4 group and three sRNA-seq libraries for the control group) were generated using the NEBNext Multiplex Small RNA Library Prep Kit for Illumina (NEB, Ipswich, MA, USA) according to the manufacturer’s protocols. Briefly, for the construction of each sRNA library, a total amount of 1 μg RNA was ligated to a 5′ and a 3′ linker sequentially according to the protocol (NEB, Ipswich, MA, USA). Next, reverse transcription was carried out using ProtoScript II Reverse Transcriptase (NEB, Ipswich, MA, USA) for 1 h at 50 °C. For miRNA library enrichment, PCR amplification with adapter-specific primers was performed using a ProtoScript^®^ II First Strand cDNA Synthesis Kit (NEB, Ipswich, MA, USA) according to the protocol, and purification was performed using 15% nondenaturing PAGE. PCR products from 140–160 bp were subjected to library preparation for sRNA sequencing analysis. Single-end reads were sequenced on an Illumina HiSeq™ 2500 System (Illumina, San Diego, CA, USA) at a read length of 50 bp, the most frequently used sequencing strategy [37].

Analysis of the miRNA sequencing data was performed as described previously [38]. Before data analysis, the raw data were subjected to an in-house program, ACGT101-miR (LC Sciences, Houston, Texas, USA), to remove adapter dimers, junk, low complexity, and repeats. Subsequently, unique sequences with lengths of 18–40 nucleotides were mapped to porcine precursors in miRbase 21.0 by BLAST search to identify known miRNAs and novel 3p- and 5p-derived miRNAs ftp://mirbase.org/pub/mirbase/ (accessed on 12 December 2019) [39]. Non-coding RNAs were annotated as rRNAs, tRNAs, small nuclear RNAs (snRNAs), and small nucleolar RNAs (snoRNAs). These RNAs were aligned and then subjected to the BLAST search against Rfam http://www.sanger.ac.uk/software/Rfam (accessed on 14 December 2019) and GenBank databases http://www.ncbi.nlm.nih.gov/genbank/ (accessed on 14 December 2019) [40].

### 2.4. Identification and Target Prediction of DEmiRNAs

We used deep sequencing to identify differentially expressed miRNAs (DEmiRNAs) between porcine GCs of the RBP4 group and the control group. MiRNA expression levels are represented as reads per million (RPM) values. The fold-change was calculated based on the RPM values. The *p*-value was adjusted by multiple hypothesis tests, and the *p*-value threshold was determined by the false discovery rate (FDR). The criteria for identifying DEmiRNAs were as follows: (1) *p*-value < 0.05 and (2) |fold change| > 2 between both groups. To understand the functional roles that the DEmiRNAs may play in porcine GCs, miRanda http://www.microrna.org/microrna/ (accessed on 20 April 2020) was used to predict their target genes. Cytoscape was used to visualize the DEmiRNAs and target genes.

### 2.5. Functional Enrichment Analysis

We employed a hierarchical clustering algorithm to cluster the DEmiRNAs. Gene enrichment and functional annotation analyses were conducted using the Gene Ontology GO, http://geneontology.org/ (accessed on 19 June 2020) and Kyoto Encyclopedia of Genes and Genomes KEGG, http://www.genome.jp/kegg (accessed on 19 June 2020) platforms [41]. Only GO terms or KEGG pathways with *p*-values < 0.05 were considered significantly enriched. All miRNA sequencing data can be accessed via the Gene Expression Omnibus (GEO) database from NCBI https://www.ncbi.nlm.nih.gov/geo (accessed on 24 July 2019). The GEO accession record is GSE134735.

### 2.6. RT-qPCR

Total RNA samples (100 ng) were reverse transcribed using the PrimeScript RT reagent Kit (Takara, Dalian, China) following standard protocols. The level of miRNA was quantified by real-time quantitative polymerase chain reaction (RT-qPCR) according to established protocols. RT-qPCR was performed by combining 10 μL SYBR (Takara, Dalian, China), 2 μL cDNA, 7 μL ddH_2_O, and 1 μL each of forward and reverse primers. The thermal cycling conditions were 15 min at 95 °C followed by 40 cycles of 10 s at 95 °C and 30 s at 60 °C. The Mx3000P system (Stratagene, San Diego, CA, USA) was used to quantify the abundance of selected genes. The primer sequences are listed in Appendix A, and the primers were synthesized in Comate Bioscience (Changchun, China). *U6* and *GAPDH* were used as internal controls to correct for miRNA and mRNA analytical variations, respectively [42]. The expression levels of chosen differentially expressed miRNAs were calculated using the delta–delta Ct (2^−ΔΔct^) method.

### 2.7. Statistical Analysis

SPSS version 17.0 (SPSS, Inc., Chicago, IL, USA) was used for statistical analysis. The experimental data are expressed as the mean ± standard error of the mean (SEM). The K–S test and Levene test were performed to check the normality and homoscedasticity of the data. Student’s *t*-test for independent samples was employed to analyze significant differences between means. * *p* < 0.05 indicated statistical significance, and ** *p* < 0.01 was considered to be a very significant difference.

## 3. Results

### 3.1. Summary of miRNA Deep-Sequencing Data

sRNA-seq was performed in biological triplicates to elucidate the regulatory mechanism of the response of GCs to *RBP4* overexpression. *RBP4* mRNA was significantly increased in the RBP4 group by 111.6-fold. In addition, RBP4 protein was overexpressed in the RBP4 group (Appendix A).

A total of approximately 91.2 Mb raw reads were obtained from the six sRNA libraries. The six libraries of the two groups yielded 45.5 M and 45.7 M raw reads. Of these, 8.2~12.9 Mb reads were retained as clean reads after eliminating reads with no 3′ adapter, reads with 5′ adapter contaminants, and reads with a poly A/poly T tail (Table 1). Averages of 9,080,992 (Control group) and 8,433,749 (RBP4 group) reads were annotated in miRBase. The sRNA-seq results revealed that the low-quality reads made up less than 0.1% of reads and that the remaining reads had a Q value ≥ 30 (>90% of reads). Altogether, the classification of these sRNAs showed that averages of 41.70% (control group) and 41.60% (RBP4 group) were previously unreported miRNAs, and rRNA and known miRNAs were also present (Figure 1). As shown in Figure 2, most sequences were 20–23 nt in length; specifically, most sequences were 22-nt, followed by 21, 23, and 24-nt, which was in accordance with the length characteristics of mature miRNAs.

The percentage represents clean reads/raw reads; perfect matches include those clean reads that completely matched the reference genome.

### 3.2. DEmiRNAs in GCs Overexpressing RBP4

Among the miRNAs identified, 2783 were known miRNAs, and 776 were novel miRNAs. In total 17 miRNAs were significantly different between the experimental and control groups (*p* < 0.05). Among the DEmiRNAs, seven were upregulated and 10 were downregulated. All the DEmiRNAs are listed in Table 2.

### 3.3. Validation of the DEmiRNAs by RT-qPCR

In the present study, all DEmiRNAs were validated by RT-qPCR. The RT-qPCR results were consistent with the sequencing data. In the RBP4 group, ssc-miR-194a-5p, ssc-miR-132, PC-5p-17154, ssc-miR-193-5p, ssc-miR-135, ssc-miR-32, and ssc-miR-98 were upregulated, while the other ten DEmiRNAs were downregulated (Figure 3).

### 3.4. Prediction and Analysis of DE miRNA Target Genes

Most miRNAs were predicted to regulate hundreds or even thousands of target genes at the same time. Appendix A shows the DEmiRNAs and their corresponding validated target genes in porcine GCs treated with RPB4. There were 4237 predicted targets for 10 downregulated DEmiRNAs and 12,316 predicted relationships. There were 2985 predicted target genes for seven upregulated DEmiRNAs and 7214 predicted relationships (Figure 4). The DEmiRNAs with the most predicted target genes were miR-1343, miR-1277, and miR-145-5p, and the number of target genes was 2688, 1581, and 1552, respectively. Many target genes showed potential regulation by multiple miRNAs. Cytochrome P450 19A1 (*CYP19A1*) and *caspase 9* were predicted to be targeted by nine and four DEmiRNAs, respectively.

To further identify candidate miRNAs involved in porcine folliculogenesis, we integrated the miRNA and mRNA sequencing data (the detailed mRNA sequencing data are shown in another published paper [35]). Based on the matching and target prediction analyses, we found that the mRNA expression of *SEMA6D*, *KIT*, and *RAB3B* was strongly correlated with the corresponding miRNAs, namely, miR-132, miR-135, and miR-32, respectively. Both the RT-qPCR and sequencing results showed that the target genes identified in the current study had expression patterns between the RBP4 and control groups that were opposite those of their respective DEmiRNAs (Figure 5).

### 3.5. GO and Functional Classification

A total of 20,787 predicted genes were classified into three functional categories: 25 biological process (BP) terms, 15 cellular component (CC) terms, and 10 molecular function (MF) terms. In the BP category, most of the annotated targets were involved in the oxidation–reduction process, regulation of transcription, DNA template-related processes, and positive regulation of transcription from the RNA polymerase II promoter. In the CC category, most targets were related to the terms membrane, integral component of membrane, and cytoplasm. In the MF category, most targets were associated with protein binding, metal ion binding, ATP binding, and nucleotide-binding (Figure 6).

The significantly enriched (*p* < 0.05) pathways are shown in Figure 7. Furthermore, KEGG pathway analysis demonstrated that the target genes were involved mainly in pathways related to metabolism, such as the insulin resistance pathway, cholesterol metabolism pathway, and fatty acid degradation signaling pathway. The target genes were also involved in pathways related to immunity, such as the chemokine signaling pathway, PPAR signaling pathway, FoxO signaling pathway, and NF-kappa B signaling pathway. Pathways related to folliculogenesis (the steroid hormone biosynthesis, oxidative phosphorylation, and AMP-activated protein kinase (AMPK) signaling pathways) were also significantly enriched.

## 4. Discussion

GCs are the most important cell type in the ovary, and they play a critical role in folliculogenesis and ovulation. Folliculogenesis is a dynamic and complex process and has been proven to be modulated by the orchestrated actions of endocrine factors, paracrine factors, and the spatiotemporal expression of several genes and miRNAs [43,44,45].

RBP4, a novel adipocytokine that was originally found to be released by the liver, has a wide range of actions in the metabolism of retinol. Recently, RBP4 has been detected in serum, plasma, liver, adipose, and kidney samples, and dysregulation of RBP4 has been associated with a variety of diseases [17,19,46,47,48], including ovarian diseases, such as PCOS and ovarian cancer in humans [17,18,49,50,51] and ovarian cysts in swine [33]. These diseases may be caused by endocrine disorders. As one of the largest classes of gene regulators, miRNAs are responsible for fine-tuning gene expression and are thought to regulate 20% to 30% of the genomes of domestic livestock [52]. Several studies have pointed out the potential role of miRNAs in GC follicular development and have suggested that aberrations in miRNA expression may play important roles in apoptosis and follicular atresia progression by regulating genes and pathways involved in ovarian diseases [53,54,55].

The current study utilized high-throughout RNA sequencing technology to characterize the global miRNA expression profiles of porcine GCs with modified RBP4 expression and GCs that were untreated. We used a lentiviral vector containing an RBP4 gene fragment for gene expression. The advantages of this method are its longer expression time, diminished cytotoxicity and immune response mediated by host cells, and strong induction of infection in primary cells such as porcine GCs. We constructed six libraries in total, comprising three from each group. More than 60.6 million clear reads were obtained. According to size distribution analysis, 22-nt RNAs were the most abundant, followed by 21 and 23-nt RNAs. The characteristics of the length distribution were consistent with the typical size of miRNAs produced by Dicer processing and were consistent with previous results [39,56]. These findings indicate that the depth and quality of our sequencing data are satisfactory. Overall, 17 miRNAs showed differential expression between the RBP4 overexpression and control groups. The RT-qPCR results were consistent with the sequencing data and indicated that porcine ovarian GCs overexpressing RBP4 exhibited altered miRNA expression. To further identify candidate miRNAs involved in porcine ovarian GCs, we integrated the miRNA and mRNA sequencing data [35]. Based on the assessment, we found three miRNA-target gene pairs, namely miR-135-*KIT*, miR-132-*SEMA6D*, and miR-32-*RAB3B*, that might participate in porcine follicular development. Compared with the DEmiRNAs, the target genes observed in the current study had opposite expression patterns between the RBP4 and control groups according to both sequencing and RT-qPCR data. These findings from our study support the post-transcriptional repressive effects of miRNAs on their target genes [57]. These DEmiRNAs might regulate the proliferation of porcine GCs by affecting *KIT*, *SEMA6D*, and *RAB3B*.

Many previous studies have shown that the above DEmiRNAs are related to folliculogenesis and ovarian diseases in different species. Hu et al. [58] found that miR-1343 could promote apoptosis and inhibit proliferation of porcine GCs by targeting transforming growth factor-beta receptor type I (*TGFBR1*). TGFBR1, also known as activin receptor-like kinase 5, is required for female reproductive tract integrity and function and that disruption of *TGFBR1*-mediated signaling leads to catastrophic structural and functional consequences in the oviduct and uterus [59]. In addition, Li et al. [60] recently showed that lower expression of miR-1343 induces apoptosis in the ovarian cancer cell line OVCAR3 by suppressing the *TGF-beta* receptor in vitro. In agreement with these findings, in the current study, we found that miR-1343 was significantly downregulated in the GCs of the RBP4 group compared to those of the control group, indicating that RBP4 modulation of miRNA expression promotes GC apoptosis. miR-148a was reported to inhibit sphingosine-1-phosphate receptor 1 (*S1PR1*) expression to suppress SKOV3 cell migration and invasion [61]. In another study, the expression of miR-193 was increased in murine ovaries after cadmium exposure via decreases in the expression of the follicle-development-related factors stem cell factor (*SCF*) and *c-kit* [62]. In PCOS, miR-145 was found to play an important role in regulating insulin receptor substrate I (*IRS1*). *IRS1* is a vital factor of the PI3K/AKT pathway that regulates ovarian function. In the present study, DEmiRNAs, such as miR-132, were found to exhibit apoptotic effects and regulate multiple cellular pathways. miR-132 has been shown to be involved in the death of HccLM3 cells, which act as a liver cancer cell model, and the expression of miR-132 progressively increases in HccLM3 cells [63]. The results of the current study motivate us to speculate that RBP4 overexpression causes cellular and molecular changes in the ovary, where miRNAs may play a central role.

In addition, KEGG pathway analyses showed that the DEmiRNAs might exert their functions through pathways related to PPARs, FoxO, NF-kappa B, AMPK signaling, oxidative phosphorylation, and steroid hormone biosynthesis. Several of the highly enriched signaling pathways have been shown to be involved in ovarian physiological functions, such as the proliferation and apoptosis of GCs and follicular development. Aberrant mitochondrial oxidative phosphorylation occurs in obese females with type 2 diabetes and insulin resistance, which are related to RBP4 [64]. Oxidative phosphorylation is also involved in GC follicular atresia and apoptosis [65]. In chicken ovaries, the AMPK signaling pathway has been reported to promote GC apoptosis via miR-204 [66]. The AMPK signaling pathway plays roles in regulating ovarian GC proliferation, steroidogenesis, and oocyte maturation [67,68,69]. The FoxO signaling pathway was predicted to be affected by downregulated miRNAs [70]. Another study recently proposed a relationship between the pleiotropic effects of diverse exogenous factors and the PPAR pathway that may underlie the modulation of lipid and glucose metabolism in two main chronic diseases, namely PCOS and nonalcoholic fatty liver disease (NAFLD) [71]. However, a functional study using the miRNA knockdown/mimic approach is required to validate the association of the miRNA-miRNA interaction and the associations of differentially expressed miRNAs with the predicted pathways.

## 5. Conclusions

In this study, we identified DEmiRNAs between porcine GCs overexpressing RBP4 and control porcine GCs by using deep sequencing. In addition, our data support the notion that several miRNAs are involved in important biological processes associated with folliculogenesis and pathogenesis. These results are useful for further studies investigating the role of RBP4 in porcine GCs.

## Figures and Tables

**Figure 1 animals-11-01391-f001:**
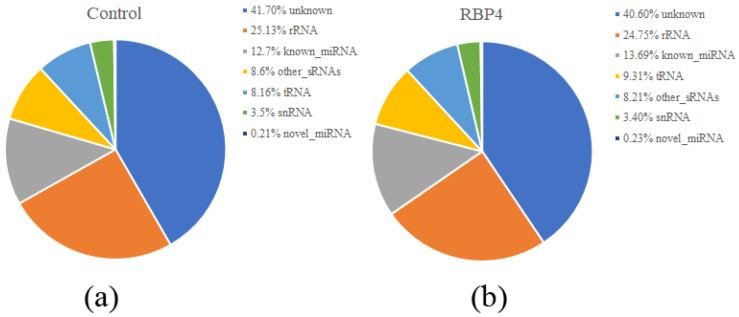
Annotation of sRNA sequences. (**a**) Classification of sRNAs of the control group; (**b**) classification of sRNAs of the RBP4 group.

**Figure 2 animals-11-01391-f002:**
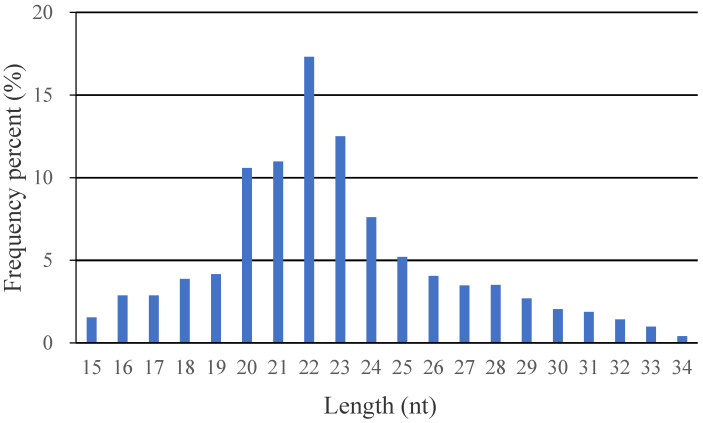
Length distribution of small miRNAs.

**Figure 3 animals-11-01391-f003:**
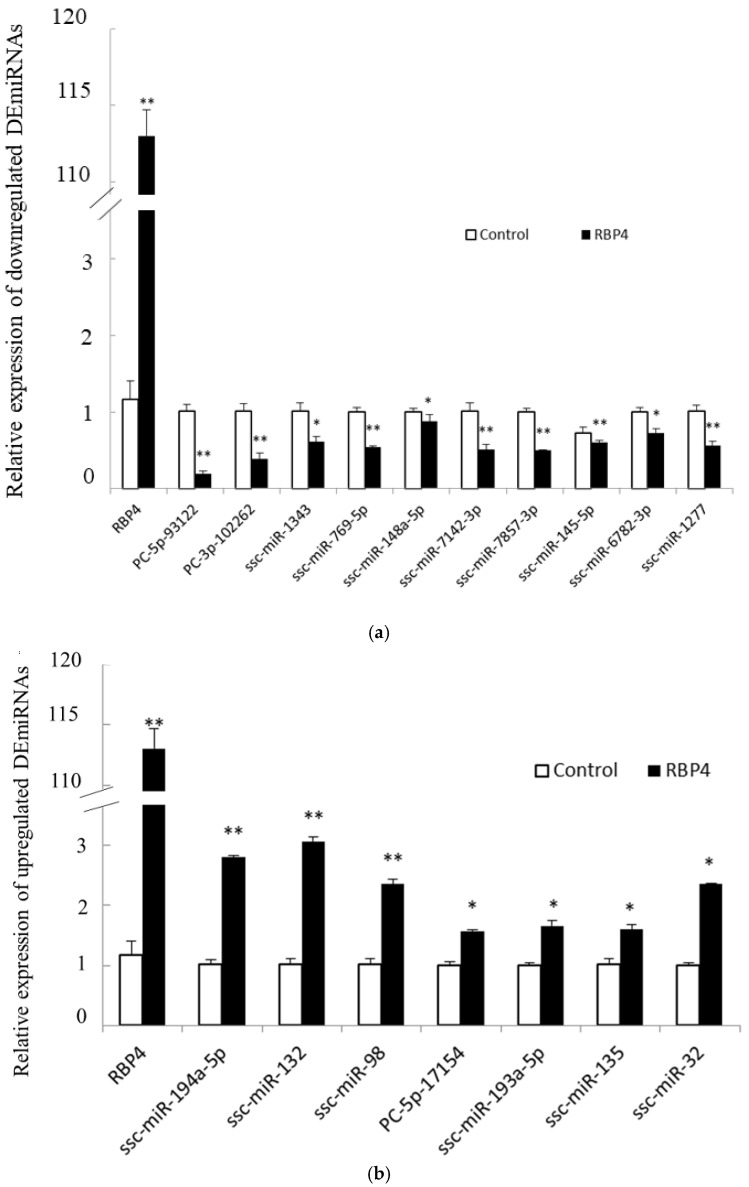
RT-qPCR validation of DEmiRNAs. (**a**) Relative expression of 10 downregulated DEmiRNAs. (**b**) Relative expression of 7 upregulated DEmiRNAs. The data are expressed as the mean ± SEM (* *p* < 0.05, ** *p* < 0.01).

**Figure 4 animals-11-01391-f004:**
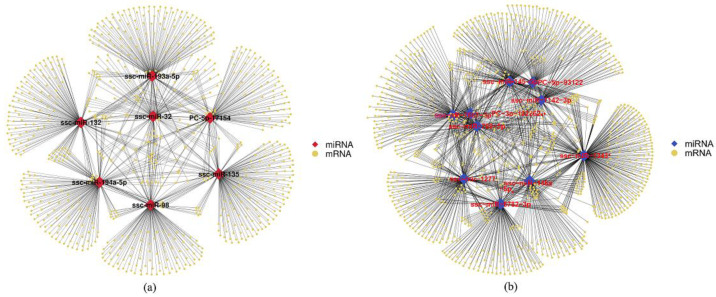
Target genes of the DEmiRNAs. (**a**) Interaction network of the upregulated DEmiRNA target genes. (**b**) Interaction network of the downregulated DEmiRNA target genes. Red blocks in the center indicate upregulated DEmiRNAs. In contrast, blue blocks indicate downregulated DEmiRNAs. Yellow blocks indicate mRNAs.

**Figure 5 animals-11-01391-f005:**
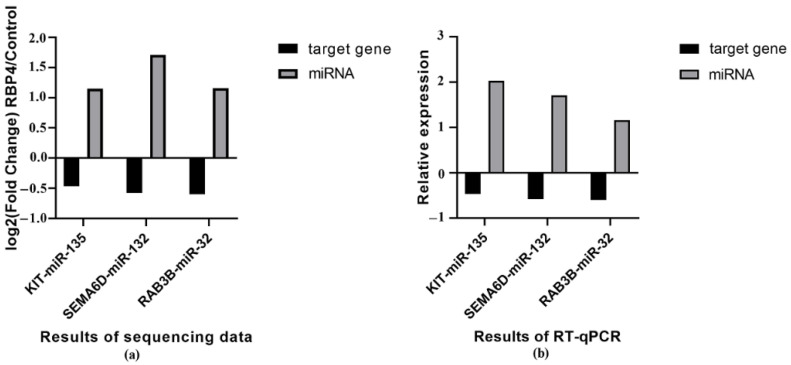
Sequencing and RT-qPCR analyses of the DEmiRNAs and their predicted target genes in the RBP4 and control groups (*n* = 3 replicates per group). (**a**) Results of sequencing. (**b**) Results of RT-qPCR.

**Figure 6 animals-11-01391-f006:**
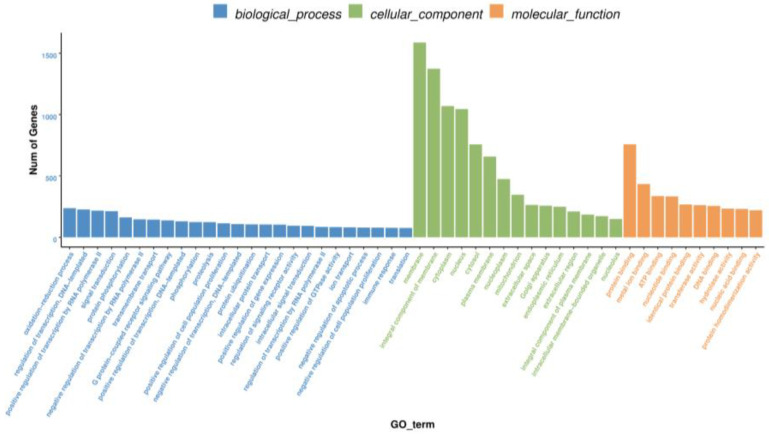
GO functional analysis of the potential targets of the DEmiRNAs.

**Figure 7 animals-11-01391-f007:**
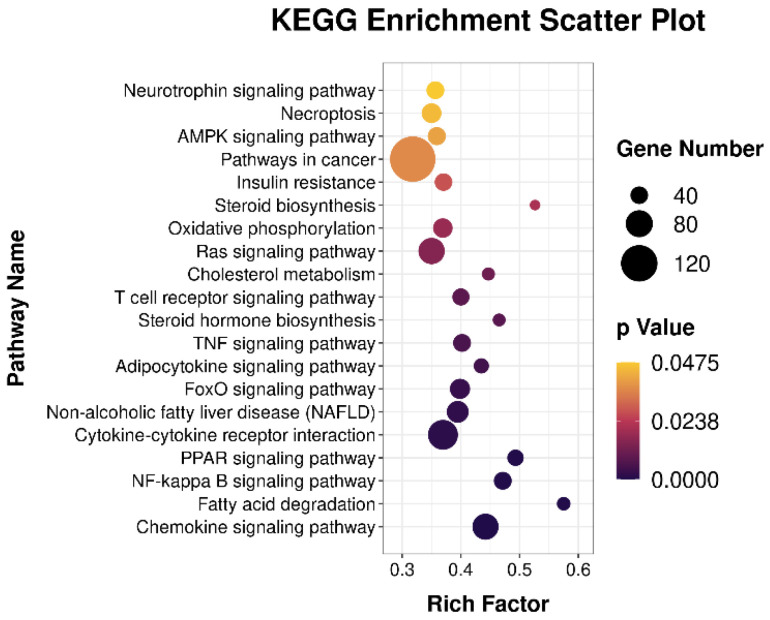
Pathways involving the predicted targets of the DEmiRNAs related to reproductive processes in GCs.

**Table 1 animals-11-01391-t001:** Statistical analysis of sequencing data from the control group and RBP4 treatment group.

Sample	Raw Reads (M)	Clean Reads (M)	Clean Ratio (%)	Clean Q30 Ratio (%)	Annotated Reads	Readsperfect Matches in miRBase (SSCROFA 10.2)
control_1	11.9	8.5	71.7	99.2	8,499,093	7,195,389
control_2	13.9	9.9	70.6	99.2	9,844,804	8,610,887
control_3	19.7	12.9	65.6	99.3	12,937,609	11,436,699
RBP4_1	17.0	10.9	64.5	99.2	10,929,835	9,398,610
RBP4_2	16.3	10.2	62.5	99.3	10,206,670	8,731,703
RBP4_3	12.4	8.2	65.8	99.2	7,608,852	7,170,935

**Table 2 animals-11-01391-t002:** Significant DEmiRNAs (*p* < 0.05).

Name of the miRNA	Fold-Change (RBP4 vs. Control)	*p*-Value
Downregulated		
PC-5p-93122	−20.50	7.67 × 10^−3^
PC-3p-102262	−18.49	3.62 × 10^−2^
ssc-miR-1343	−4.89	2.91 × 10^−2^
ssc-miR-769-5p	−3.86	3.60 × 10^−2^
ssc-miR-148a-5p	−3.77	4.81 × 10^−2^
ssc-miR-7142-3p	−3.76	1.86 × 10^−2^
ssc-miR-7857-3p	−2.70	4.02 × 10^−2^
ssc-miR-145-5p	−2.66	3.26 × 10^−2^
ssc-miR-6782-3p	−2.56	2.29 × 10^−2^
ssc-miR-1277	−2.17	3.62 × 10^−2^
Upregulated		
ssc-miR-194a-5p	4.45	1.67 × 10^−2^
ssc-miR-132	3.27	2.47 × 10^−2^
ssc-miR-98	3.16	3.32 × 10^−2^
PC-5p-17154	2.50	2.87 × 10^−2^
ssc-miR-193a-5p	2.47	3.65 × 10^−2^
ssc-miR-135	2.22	1.51 × 10^−2^
ssc-miR-32	2.23	3.21 × 10^−2^

## Data Availability

Data are contained within the article and Appendix A. Raw data are available on request from the corresponding author.

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
