# Peer review of "The Effect of RBP4 on microRNA Expression Profiles in Porcine Granulosa Cells"

_animals, 2021, doi:10.3390/ani11051391_

Round 1
Reviewer 1 Report
The authors performed a good work and the experimental approach and the results obtained deserve publication. However, I think that the manuscript would benefit of an extensive revision of the English grammar which facilitates the reading.
Reviewer 2 Report
This study highlighted the effects of retinol binding protein 4 on gene expression via microRNA.
The work is very substantial, uses a highly specialized technique, the results are well presented and the discussion is consistent with the results.
In light of this, the work is suitable for publication in the journal after a simple revision.
When inserting a virus with a particular gene to be overexpressed, the problem of the possible pharmacological effect arises, all this should be commented on in the discussion.
Minor
Line 83: Sun et al. [33]
Line 84: delete [33]
Lines 91-93: delete “To our……with RBP4”
Line 124: cite the study or detail the method
Lines 193-196: delete “In the…..data”
Line 197: add legend to Table S1
Lines 222-223: delete “DEmiRNAs……data”
Lines 238-239: delete “Identification……microrna)”
Line 290: delete Authors
Line 309: “throughout”
Line 331: Hu et al. [58]
Line 333: delete [58]
Line 333: Lin et al. [59]
Line 335: [59]
Figure and legend: figures and legends need to be further apart from the text, add space before and after
Reviewer 3 Report
Dear author,
The title of the manuscript augured an interesting study on the possible effects of RBP4 on the gene expression patterns of porcine granulosa cells. However, the methods used are very difficult to understand since, although the authors state that they analyze the profiles of small RNAs in their results, the manuscript does not explain how they have separated these small RNAs and, specifically, the microRNAs from the total RNA extracted. . In addition, the manuscript presents many typographical errors, such as abbreviations that do not indicate what they are (lines 10,13, 32, among others), and excess words (for example, line 290). Furthermore, the authors use italics interchangeably when referring to gene expression and when not (lines 70 to 93). On the other hand, in line 268 they speak of genes when they should specify whether they refer to mRNAs, or microRNAs. But the biggest impediment to its publication lies in the material and methods section, where the authors do not follow a chronological order on the methods used (first they explain how they have made the microRNA libraries and then how they have extracted the total RNA), they do not explain how they have separated the different types of small RNAs, they use a parametric test without previously checking the normality and homoscedasticity of the data, and they refer to published works but do not add the references of these works.
Reviewer 4 Report
Granulosa cells (GCs) are important for follicles development both on morphological and endocrinological aspect. The researchers of this manuscript tried to analysis the biological path of the RBP4 functioned in GCs by overexpress RBP4 in in vitro cultured GCs and miRNA-Seq. They found 10 downregulated and 7 upregulated miRNAs in the overexpressed RBP4 GCs compared with control group, then they did Gene Ontology and KEGG enrichment analyses of these miRNAs. This manuscript provided interesting data for researchers in the field of follicle developmental regulation.
Comments
- There are 10 downregulated miRNAs and 7 upregulated miRNAs, why the authors did not validate all of them, especially the first two downregulated miRNAs?
- The discussion is not well organized. Most of the content repeats the introduction and the results, the authors should pay much more attention on how to understand the significance of the results.
- The authors marked “(a)”and “(b)”in the figures of Figure 5, but in the figure legends the authors instead of “(a)”and “(b)”with “(A)”and “(B)”.
Round 2
Reviewer 1 Report
the authors did a good job revising the English grammar
Reviewer 3 Report
Dear authors,
The manuscript has been improved, but some changes are necessary yet.
- Line 207: the name of genes must be written in italics.
- Line 207: the authors should indicate references that use these housekeeping in similar samples.
